

# Light wavelength and pulsing frequency affect avoidance responses of Canada geese

Ryan Lunn[1], Patrice E. Baumhardt[1], Bradley F. Blackwell[2], Jean Paul Freyssinier[3] and Esteban Fernández-Juricic[1]

[1] Department of Biological Sciences, Purdue University, West Lafayette, IN, United States of America
[2] United States Department of Agriculture, Animal and Plant Health and Inspection Services, National Wildlife Research Center, Sandusky, OH, United States of America
[3] Lighting Research Center, Rensselaer Polytechnic Institute, Troy, NY, United States of America

## ABSTRACT

Collisions between birds and aircraft cause bird mortality, economic damage, and aviation safety hazards. One proposed solution to increasing the distance at which birds detect and move away from an approaching aircraft, ultimately mitigating the probability of collision, is through onboard lighting systems. Lights in vehicles have been shown to lead to earlier reactions in some bird species but they could also generate attraction, potentially increasing the probability of collision. Using information on the visual system of the Canada goose (*Branta canadensis*), we developed light stimuli of high chromatic contrast to their eyes. We then conducted a controlled behavioral experiment (*i.e.*, single-choice test) to assess the avoidance or attraction responses of Canada geese to LED lights of different wavelengths (blue, 483 nm; red, 631 nm) and pulsing frequencies (steady, pulsing at 2 Hz). Overall, Canada geese tended to avoid the blue light and move towards the red light; however, these responses depended heavily on light exposure order. At the beginning of the experiment, geese tended to avoid the red light. After further exposure the birds developed an attraction to the red light, consistent with the mere exposure effect. The response to the blue light generally followed a U-shape relationship (avoidance, attraction, avoidance) with increasing number of exposures, again consistent with the mere exposure effect, but followed by the satiation effect. Lights pulsing at 2 Hz enhanced avoidance responses under high ambient light conditions; whereas steady lights enhanced avoidance responses under dim ambient light conditions. Our results have implications for the design of lighting systems aimed at mitigating collisions between birds and human objects. LED lights in the blue portion of the spectrum are good candidates for deterrents and lights in the red portion of the spectrum may be counterproductive given the attraction effects with increasing exposure. Additionally, consideration should be given to systems that automatically modify pulsing of the light depending on ambient light intensity to enhance avoidance.

Corresponding author
Ryan Lunn, rlunn@purdue.edu

## INTRODUCTION

Globally, avian populations are declining (*Lees et al., 2022*). Estimates from both North America and Europe, respectively, reported decreases in avian abundance of 27–30% (1970–2017; *Rosenberg et al., 2019*) and 17–19% (1980–2018; *Burns et al., 2021*) due to different anthropogenic sources (*Lees et al., 2022*; *Loss, Will & Marra, 2015*). The third highest anthropogenic source of direct avian mortality is collisions with vehicles, behind cat predation and collisions with buildings. In the US alone, vehicles are estimated to annually kill between 88.7 to 339.8 million individuals (*Loss, Will & Marra, 2014*; *Loss, Will & Marra, 2015*). A subset of those vehicle collisions includes collisions between aircraft and birds, hereafter bird strikes, which occur around the globe (*Australian Transportation Safety Bureau, 2019*; *Dolbeer et al., 2023*; *Sarkheil, Talaeian Eraghi & Vatan Khah, 2021*). Besides the loss of birds, bird strikes cause substantial economic damage and pose a major safety hazard to aviation (*Allan, 2000*; *DeVault et al., 2018*; *Dolbeer et al., 2023*). The estimated annual cost of bird strikes is $205 million dollars in the U.S, and $1.2 billion dollars globally (*Allan, 2000*; *Dolbeer et al., 2023*). Additionally, over a 31-year period, bird strikes have been the cause of the destruction of 271 aircraft and 292 human fatalities (*Dolbeer et al., 2023*).

Airport wildlife management programs aim to mitigate the risk of bird strikes, but are limited to the spatial jurisdiction of the airfield and the immediate airport surroundings (*Blackwell et al., 2009a*; *Blackwell et al., 2012*; *Dolbeer, 2011*). There are no specific bird-deterrence practices in place outside of the airport property (*Federal Aviation Administration, 2018*). An idea originally proposed in the 1970's (*Lustick, 1973*; *Larkin et al., 1975*) that gained more attention in recent decades (*Blackwell, 2002*; *Blackwell & Fernández-Juricic, 2013*) to help minimize the risk of bird strikes is the use of onboard lighting systems (*Blackwell & Bernhardt, 2004*). In principle, onboard lighting could increase the distance at which birds first detect and draw their attention to an approaching aircraft (*Blackwell et al., 2009b*; *Blackwell et al., 2012*; *Blackwell & Fernández-Juricic, 2013*). The increase in detection distance would provide more time for the animal to enact an avoidance response and if the object is perceived as threatening provoke a relatively longer escape distance (*i.e.,* flight initiation distance (FID)), ultimately reducing the probability of collision (*Blackwell et al., 2009b*; *Blackwell et al., 2012*; *Doppler et al., 2015*). Typically, cues that animals perceive to be threatening, from an antipredator theory perspective (*Caro, 2005*) include direct approach (*i.e.,* a collision course), fast approach speed, and object size (*Stankowich & Blumstein, 2005*). However, the application of antipredator theory to predict behavioral responses should be used cautiously (*Lunn et al., 2022*). For example, a fast-moving vehicle might not be perceived as the same amount of risk as a fast-moving predator (see also *DeVault et al., 2015*).

Special consideration is required when developing visual stimuli such as lights to stimulate the avian visual system, as opposed to the human visual system. Birds visually perceive their world differently from humans (*Cuthill et al., 2006*), with substantial variation among different bird species (*Hart, 2001*; *Hart & Hunt, 2007*; *Dolan & Fernández-Juricic, 2010*). Therefore, it is necessary to understand the visual sensory and cognitive

perspectives of the target species to establish (a) the range over which the visual stimulus is not only detectable but salient enough to elicit a behavioral response, and (b) that the behavioral response aligns with the management goal (*i.e.,* the light stimulus leads to avoidance behavior instead of attraction behavior or no response) (*Blackwell & Fernández-Juricic, 2013*; *Elmer et al., 2021*; *Fernández-Juricic, 2016*). Mathematical models that utilize specific properties of the visual system of the target species can emulate the processing of visual stimuli in the sensory system (*e.g.,* receptor-noise limited model, visual acuity estimates) allowing us to estimate detection distance or stimulus saliency (*Pettigrew et al., 1988*; *Vorobyev et al., 2001*; *Vorobyev & Osorio, 1998*). These models have yielded the distances at which objects of a certain size could be initially resolved (*Tisdale & Fernández-Juricic, 2009*; *Tyrrell et al., 2013*) as well as specific wavelengths of light that would tend to stimulate the visual system more relative to the environmental background (*Doppler et al., 2015*; *Goller et al., 2018*). Both of which potentially affect animal decision-making. Standardized behavioral assays that quantitatively measure avoidance/attraction responses are necessary to explicitly evaluate whether responses to candidate lights indeed lead to avoidance behavior (*Blackwell et al., 2009a*; *Blackwell et al., 2009b*; *Doppler et al., 2015*; *Goller et al., 2018*; *Goller et al., 2018*). For instance, *Goller et al. (2018)* found that of five different candidate LED lights with high levels of visual stimulation, only blue (464 nm) and red lights (633 nm) caused avoidance behavior in the Brown-headed cowbird (*Molothrus ater*).

Standardized behavioral assays offer some benefits in the process of developing novel stimuli for avian deterrence purposes. First, these assays allow for the serial control of multiple confounding factors (*i.e.,* satiation levels, body condition, ambient light, identity of individuals, *etc.*) that could influence behavioral responses. Controlled conditions are essential to narrow down the basic behavioral response to the stimulus before establishing whether such a response is augmented in the presences of other confounding factors. This behavioral assay process is necessary to conclude whether the chosen stimulus can be effective under different environmental and ecological conditions (*Dominoni et al., 2020*; *Elmer et al., 2021*; *Emerson et al., 2022*). Second, standardized behavioral assays provide the opportunity to examine the existence of habituation or sensitization to treatments *via* repeated exposure to the same individuals (*Blumstein, 2016*; *Rankin et al., 2009*). If a stimulus generates avoidance responses upon the first exposure, but that response extinguishes over repeated exposures, leading to an insufficient response or no response at all, continued development of new stimuli related technology might not be cost-effective. Third, standardized behavioral assays can be used for multiple, rapid evidence-based tests of different stimuli to expedite the development of avian deterrents (*Goller et al., 2018*; *Thady, Emerson & Swaddle, 2022*). Fourth, standardized behavioral assays allow for the quantification of the probability of avoidance to the stimuli, which can be used to inform modeling approaches to estimate the relative risk of bird strikes given different stimuli treatments (*Ghazaoui et al., 2023*; *Lunn et al., 2022*).

The goal of our study was to evaluate behavioral responses of Canada geese (*Branta canadensis*) to light stimuli that are visually salient to their eyes. To date, lights of high chromatic contrast to the Brown-headed cowbird's visual system have been shown to both

incite avoidance responses and enhance the distance animals become aware of approaching vehicles (*Doppler et al., 2015*; *Goller et al., 2018*). However, several studies have shown bird attraction to different light sources (*Poot et al., 2008*; *Reed, Sincock & Hailman, 1985*; *Syposz et al., 2021*). If birds are attracted to light stimuli (*i.e.,* moving towards the light) then lights on aircraft might actually increase the probability of collision. We set out to explicitly test behaviorally the avoidance or attraction response of Canada geese to lights of high chromatic contrast relative to their visual system in a standardized behavioral experiment using a single-choice test.

We chose this species because (1) bird strikes involving geese are particularly costly and (2) pose a substantial threat to the safety of the aircraft and ultimately its passengers (*DeVault et al., 2018*; *Dolbeer et al., 2023*). We used a visual contrast model (*Vorobyev & Osorio, 1998*) to choose two lights with wavelengths of high chromatic contrast to the visual system of Canada geese. Additionally, we decided to test steady and pulsing lights at 2 Hz based on previous evidence that variations in the light pulsing frequency can influence detection and escape responses in birds (*Blackwell et al., 2009b*; *Blackwell et al., 2012*; *Doppler et al., 2015*). We were interested in the effects of light wavelength, pulsing frequency, and their interaction. We used a repeated measures design that allowed us to test individual responses upon repeated exposure to different light treatments.

We measured the following behavioral responses of Canada geese: probability of avoiding the light, latency to respond to the light, and the rate of change in both head and body orientation before making a choice. The latency to respond can have bearing on how fast animals can engage in avoidance maneuvers when confronted with an approaching threat. Head orientation changes are a proxy for how an animal allocates visual attention to a given stimuli (*Dawkins, 2002*; *Fernández-Juricic & Kowalski, 2011*). Metrics of visual attention have implications for how geese visually explore lights of different wavelength and pulsing frequency. Additionally, animals might adjust body orientation to either gather information or alter their path trajectory in response to a stimulus (*Fernández-Juricic, Erichsen & Kacelnik, 2004*; *Fernández-Juricic & Kowalski, 2011*; *Gatesy & Biewener, 1991*; *Kaby & Lind, 2003*). Given that our experiment was exploratory, we had no a-priori predictions about how Canada goose behavior would change in response to our different lighting treatments.

## MATERIALS & METHODS

### Preprint text

We deposited an initial version of this manuscript in the EcoEvoRxiv preprint server (*Lunn et al., 2023*). As a result this manuscript, which has been peer reviewed, shares a large proportion of text with its preprint predecessor. The preprint can be accessed with the following link: https://doi.org/10.32942/X23029.

### Overview

We conducted our experiment in semi-natural conditions (*i.e.,* an enclosed experimental arena outdoors) at Purdue University's Ross Biological Reserve (40°24′35.16″N, 87°4′9.71″W). We ran the trials over the course of 11 different days from December

17th 2020 to January 19th 2021, outside of the migratory season (*Tacha et al., 1991*; *Wege & Raveling, 1984*), between 9:30 am and 5:00 pm.

## Animal husbandry

We used 23 Canada geese collected from Marion County, IN, that were designated for euthanasia as part of the state of Indiana's Canada geese Management program (*Indiana Department of Natural Resources, 2021*). Individual geese were identified with a randomized combination of colored leg bands (size 14 plastic bandettes; National Band & Tag Company, https://www.nationalband.com/) and a single numbered leg band. We housed the geese outdoors at the Ross Biological Reserve in a 6.10-m wide × 10.67-m long × 2.44-m tall outdoor enclosure with *ad libitum* water and food (cracked corn and Purina™ gamebird maintenance chow). We also provided pools of water for enrichment and bathing purposes. The geese were also provided with string attached to the walls of the aviary which served as pecking distractors and additional enrichment. We euthanized animals in the event of serious bodily injury or illness (*i.e.,* 24 h or more of inactivity) *via* lethal injection with a 1 mL/4.5 kg dose of Beuthanasia. No animals were euthanized as a result of our study. Upon conclusion of the experiment the animals were retained to be used as subjects for future behavioral experiments. Our experimental procedures were approved by the Institutional Animal Care and Use Committee at Purdue University (IACUC# 1401001019).

## Experimental arena

Following *Goller et al. (2018)*, we used a single-choice test experimental design, also known as a "no choice" test (*Dougherty, 2020*; *Rosenthal, 2017*), to explicitly evaluate the avoidance response of Canada geese to light stimuli of different peak wavelengths and pulsing frequencies. Single-choice tests are common in the mate-choice literature and similar in concept to a 'T' and 'Y' maze where in a symmetrical arena a single individual is exposed to a single stimulus on one side of the arena, such as a potential live mate or audio recordings of a potential mate (*D'Isa, Comi & Leocani, 2021*; *Dougherty, 2020*; *Rosenthal, 2017*; *Wagner, 1998*). Behavioral responses to the stimulus, such as latency to approach, direction of movement, duration of attention, copulation displays, avoidance, *etc.* are often used as criteria to assess attraction to the stimulus (*Amdam, & Hovland, 2011*; *Ronald, Fernández-Juricic & Lucas, 2012*; *Wagner, 1998*; *Yorzinski et al., 2013*).

Our single-choice test consisted of releasing a single Canada goose into an arena with a light stimulus on one side and an inoperable light panel on the other side. As individuals moved through the arena, they eventually reached a partition that split the pathway into a left and right side forcing individuals to make a directional choice either towards or away from the light stimulus (Fig. 1A). We used this directional choice as a proxy to establish attraction or avoidance responses to the light. When approached by threats such as high-speed aircraft, animals are often forced to make a directional responses in attempting to escape, which has potential implications for whether a collision occurs (*Bernhardt et al., 2010*).

The arena was oriented so that as the individual birds moved through the arena they moved from west to east (Fig. 1A). The experimental arena was 9.76 m long, 3.66 m wide

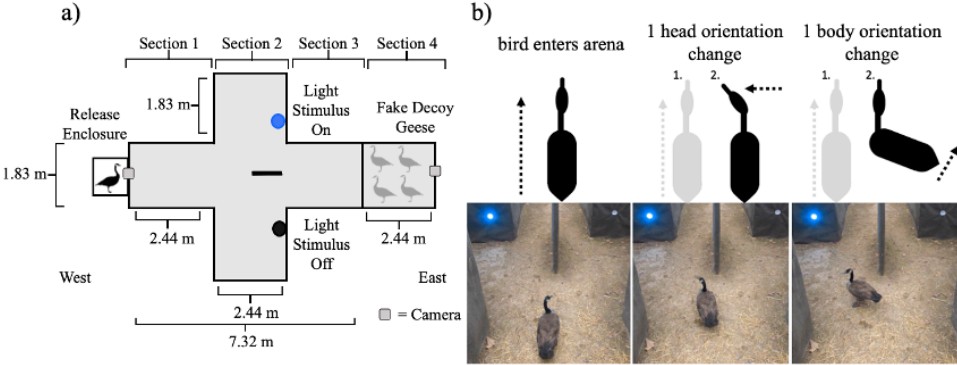

**Figure 1** **Diagrams of the behavioral experiment.** (A) Schematics of the single choice preference test arena used in this experiment. (B) A schematic representation of the behaviors noted when measuring head orientation and body orientation accompanied with sequential pictures of actual changes in head and body movement from a goose that was exposed to a blue light stimulus.

at the largest width, and 2.44 m tall throughout and was built on level ground in a forest clearing. The walls of the arena were constructed from 1.27 cm pressure treated plywood sheathing. The sides of the arena were covered in DuraWeb Geotextile landscape fabric. The top of the arena was covered with two layers of Polar Plastics multi-purpose 4-mil clear poly plastic sheeting to make the top of the arena visually homogeneous while still allowing light to enter into the arena. The arena had four different sections. The animal started in the release enclosure (61 cm × 61 cm × 61 cm) which had a wooden frame covered in 1.27 cm galvanized hardware cloth and then wrapped completely in DuraWeb Geotextile landscape fabric to prevent the animal from having visual access to the arena prior to being released.

We placed the animal inside of the release enclosure prior to the trial to provide time for the animal to acclimate (2–3 mins). The opening of the release enclosure was then moved into place alongside section 1 of the arena (Fig. 1A). The release enclosure was positioned exactly 61 cm from the walls in section 1 to standardize the position of the animal's entrance into the arena and minimize the possibility of side bias. Section 1 was 2.44 m long and 1.83 m wide, within which the animal was free to move throughout. Our protocol included removing any bird that failed to become calm or spent time probing the enclosure for escape.

As the animal moved east, away from the release enclosure into the arena, it eventually moved into section 2. In section 2, the width of the arena doubled to 3.66 m, with the length of 2.44 m remaining consistent with section 1. At 1.22 m into section 2, a partition forced the animal to move either to the left or right side within section 2. The partition was constructed of a single piece of plywood sheathing upheld on each end with a t-post (1.22 m by 2.44 m by 1.27 cm). Both ends were covered with a foam pool noodle to prevent injury in the instance an animal collided with the partition. The entirety of the partition was also wrapped in Duraweb Geotextile landscape fabric.

Both the left and right sides of section 2 were identical in width (1.83 m) to section 1. In section 2, only a single side of the partition contained a treatment light stimulus that was on and emitting light for any given trial. In the opposite chamber a lighting unit of the exact same size was visible but turned off (*i.e.,* not emitting light). The light stimuli were placed at a height of 61 cm, approximately eye level with a goose, and 1.36 m away from the center of section 2 (*i.e.,* the partition; Fig. 1A). The animal was allowed to keep moving past the partition and into a third section where both the left and right side of section 2 conjoined. Section 3 of the arena was identical in width and length as section 1. Typically, we recaptured animals in section 3.

The arena extended into section 4, which was 2.44 m long and 1.83 m wide the same width and length as section 1 and 3 (Fig. 1A). However, geese were blocked from moving into section 4 by 1.27 cm heavy duty deer fencing (*i.e.,* black square netting) staked to the ground. In section 4, we symmetrically placed four Canada goose decoys (Fig. 1A) that were visible to the live individuals in the arena. The purpose of these decoys was to draw the attention of the live individuals towards the back of the experimental arena. The decoy geese were positioned to be symmetrical on both the left and right side of section 4. The decoys were aligned so that they would directly face each other with their tail feathers pointing towards the walls of the experimental arena. The viewpoint looking toward the east side of the arena was two geese in a head down position facing each other with two geese in a head up position behind them, again facing each other.

## Behavioral experiment

Before the initiation of a trial, a Canada goose was captured in the housing enclosure and then transported on foot by the observer (RL) to the experimental arena and placed inside of the release enclosure. After placing the release enclosure into the experimental arena, the observer (RL) gently lifted the back of the release enclosure tipping it forward and patting the bottom to prompt the goose to move into the experimental arena. Prompting was necessary because during pilot trials birds tended to stay inside of the release enclosure (see also *Blackwell et al., 2019*). Once the animal walked into the experimental arena, the trial would officially start. Trials were recorded with two different GoPro Hero 7 cameras, recording at 60 frames per second, at both the west and east end of the arena (Fig. 1A). A trial concluded the moment the goose's beak entered into one of the two sides of section 2 created by the partition. Specifically, at this point the bird would no longer have direct visual access to the opposite side of the arena (Fig. 1A). Once the animal made a choice, the observer entered the arena to retrieve the animal and take it back to the holding enclosure.

In each trial, an individual was given a maximum of 10 min to make a choice. If a choice was not made after 10 min the trial stopped and the animal was retrieved and returned to the holding enclosure. Such instances were considered as mistrials, and no further measurements were taken. If an individual failed to make a choice (*i.e.,* a mistrial) three consecutive times, the individual was removed from the study. Overall, 19 out of 23 birds completed all eight treatments used in the experiment.

We utilized a repeated measures design where each individual bird was exposed to all treatment combinations. We simultaneously manipulated light color and pulsing

frequency, yielding four treatments: blue & steady, blue & pulsing (see below), red & steady, and red & pulsing. To avoid the potential confounding effects of applying a treatment combination only on the right or left sides of the arena, we exposed each individual to all four combinations of treatments on both left and the right sides of the arena for a total of eight trials. We designed the experiment so that the light would be a neutral stimulus to avoid confounding the behavioral responses with the presence of either a reward (*i.e.,* a positive stimuli) or a consequence (*i.e.,* negative stimuli).

Each individual received only one trial per day. We ensured that for the first four trials, each individual was exposed to each combination of light color and pulsing frequency. We randomized the exposure order of the light color and pulsing frequency treatment combinations as well as the light "on" side in the arena (right, left) for each individual. In the second set of four trials (trials 5–8), we again randomized the order of the color and pulsing frequency combinations, but this time with the opposite light position at which each individual was exposed to in the first four trials. Repeatedly exposing each individual to the stimulus in question was important to assess whether the light stimulus elicits a consistent response over time. An effective and non-lethal avian deterrent, such as an external light stimulus on an aircraft, would require the target species to routinely respond to the stimulus despite repeated exposures (*i.e.,* avoiding habituation) (*Blumstein, 2016*; *Lunn et al., 2022*; *Rankin et al., 2009*).

At the conclusion of each trial, we measured confounding environmental variables:: time of the day, ambient light intensity (lux, *via* Lux Light Meter Pro app; https://apps. apple.com/us/app/lux-light-meter-pro/id1292598866?platform=iphone), and temperature. We recorded time of day prior to the start of the recording of each trial. We corroborated the lux measurements with the TekPower LX1330B light meter (Kaito Electronics, Inc., Montclair, CA, USA) and decided to use the app out of logistical convenience. We measured ambient light intensity measurements directly above the housing unit of the light stimuli on both the left and right sides of the arena. We recorded temperature in Celsius with a Kestrel 3500 weather meter directly at the center of the experimental arena in section 2 at the start of the partition 1.21 m above the ground.

## Visual modelling and the light stimulus

Before the behavioral experiment and light stimulus were built, we systematically modelled the visual contrast of different LED lights based on species-specific visual properties of the Canada goose available from the literature (*Fernández-Juricic et al. 2011b*; *Moore et al., 2012*) to determine both the number and peak wavelength of the LED treatments. Using the *Vorobyev & Osorio (1998)* receptor noise limited model in the R package *pavo* (*Maia et al., 2019*), we estimated the chromatic contrast in units of JND or just noticeable differences between 201 simulated LEDs and a sky background under an ideal illuminant. The 201 simulated LED spectra were created by using the spectrum of a green (525 nm peak) LED from SuperBrightLEDs, Inc. (St. Louis Missouri, USA), then normalizing each spectral distribution to 4,000 photon counts, and shifting the peak in 2 nm intervals to produce different spectra from 300 to 700 nm.

This visual modeling exercise required (1) the spectrum of the sky to use as a background to compare the LED spectra against and (2) visual system parameters from a Canada goose. Firstly, we measured the radiance of the sky at noon on a clear day (<10% cloud cover; March 21st, 2015) and a cloudy day (>80% cloud cover; March 19th, 2015) in an open grassy field in West Lafayette, Indiana (40.417815 N, -86.942034 W) outside of the Purdue University Airport using an Ocean Insight Inc. (Orlando, FL, USA) Jaz spectroradiometer. Using a R200-7-SR reflectance probe held at 45° above ground level, we took 10 measurements of the sky (subsequently averaged); two measurements in each of the four cardinal directions and two directly up at the sky at an integration time of 30 ms. We chose the clear noon time of day as our sky background because (1) it coincided with the typical time of our behavioral experiments and (2) in bright, ambient light conditions birds rely on photopic vision, which is primarily associated with color vison and chromatic contrast (*Hart, 2001*). Secondly, we used information on the visual system of the Canada goose from (*Moore et al., 2012*). Specifically, we used the peak sensitivity of single cone photoreceptor visual pigments, absorbance of the oil droplets contained in these photoreceptors, and the relative photoreceptor density for each single cone type.

The transmittance of the ocular media for the Canada goose is not known in the literature, so in order to accurately model this, we measured the ocular media transmittance of an individual Canada goose. We measured the ocular media transmittance, following *Fernández-Juricic et al. (2019)*, by enucleating the right and left eyes and removing a small portion of the sclera at the back of the eye approximately the size of the cornea (15.7 mm). Each eye was then placed onto a custom eye holder, containing phosphate buffered saline, and 20 measurements of percent transmittance taken using an Ocean Insight Inc. Jaz spectroradiometer. The measurements from each eye were averaged together, normalized to 1, and the wavelength at 50% of the light transmitted measured ($\lambda_{T0.5}$; 369 nm). We then fitted a curve to the data using TableCurve2D v4 (Systat Software, San Jose, CA, USA; $R^2 = 0.999$) so that any noise in the spectrum below 369nm would not influence the contrast calculation results (*Fernández-Juricic et al., 2019*).

Based on visual modeling (Fig. S1), we chose two peak wavelengths of high chromatic contrast to the Canada goose visual system: LED lights with a peak at 483 nm (hereafter, blue light) and at 631 nm (hereafter, red light). We selected these specific peak wavelengths because they were (1) within each of the relative peaks of chromatic contrast and (2) readily commercially available (Fig. S1). These wavelengths were then used to build the light stimulus specifically for this behavioral experiment. The light stimulus comprised two LED arrays. The specifications and spectral distribution of the light stimulus are provided in Supplemental Information 1. We acknowledge that the specific chromatic contrasts for both the blue and red stimuli could have changed to some degree when viewed within the experimental arena as the lighting conditions varied over the course of the experiment (*i.e.*, clear *vs* cloudy). However, when we modeled these differences in clear and cloudy ambient light and sky backgrounds, we found that the contrast values were both less than a 2 JND difference at both 483 and 631 nm, respectively, with the trends of highest contrast in the blue and red wavelengths remaining the same (Fig. S1).

The light stimulus had four different light intensities for both the blue (20, 40, 80, 120 candelas) and red light (40, 80, 120, 240 candelas). However, the candela is a photometric unit of the perceived stimulus intensity (*i.e.,* radiant intensity (mW/cm$^2$)) based on sensitivity of the human visual system. Perceived intensity in humans in bright ambient conditions is related to the relative stimulation of the medium- and long-wavelength sensitive photoreceptors (*Osorio & Vorobyev, 2005*; *Sharpe et al., 2005*). In contrast, the sensation of intensity for birds in bright ambient conditions is thought to be related to the relative stimulation of the double cones, cells which are more sensitive to longer wavelengths (*Goldsmith & Butler, 2005*). Because we were interested in behavioral responses to lights of different wavelengths of high chromatic contrast, given our visual models, not perceived achromatic intensity, we controlled for the absolute stimulus intensity (*i.e.,* radiance) by selecting light intensities for each color whose peak outputs at each wavelength were radiometrically similar. In other words, the number of photons that each light produced was similar between color treatments; only wavelength and pulse differed.

We selected the blue light at 80 cd (16,159 photons/cm$^2$ at 483 nm) and red light at 120 cd (18,056 photons/cm$^2$ at 631 nm), as we wanted a sufficient light intensity that could be resolved by the geese and for which the peak output was radiometrically similar (Table S1). The total radiant intensity for the blue light stimulus was 1,315,687 (photon counts per 1,000 μs) where the radiant light intensity for the red light stimulus was 1,263,374 (photon counts per 1,000 μs). A table of the radiometric intensities at peak wavelength and total radiometric output can be found in Table S1. Furthermore, a comprehensive guide to the different units and instruments we used to measure ambient light and the light produced by the LED stimulus can be found in Table S2.

We chose two light pulsing frequencies for use in the behavioral experiment: a steady light and a light pulsing at 2 Hz. We used a steady light, as it appears to humans (>60 Hz), because it is the standard used for guiding visual flight in aviation (*Federal Aviation Administration, 2014*; *Emoto & Sugawara, 2012*). We used a 2 Hz pulsing frequency because it is within the range of safe lights for civil aviation, as pilots reported flicker vertigo when exposed to pulsing frequencies between 4 Hz and 20 Hz (*Rash, 2004*). Previous studies have shown that a light stimulus pulsing at 2 Hz was sufficient at increasing the distance a Canada goose responds to an approaching vehicle (*Blackwell et al., 2012*). Preliminary analysis of data related to the temporal visual resolution of Canada geese suggests that the steady light treatment appeared as a steady and consistent light on and the pulsing treatment appeared flashing in a light on then light off pattern (E Fernández-Juricic et al., 2020, pers. obs.). The light specifications involving pulsing rate can be found in the Supplemental Information 1.

## Potential side bias

Choice tests can be subject to side biases, that is subjects preferring to favor one side of the arena over another due to reasons not related to the stimulus in question (*Dougherty, 2020*; *Rosenthal, 2017*). Prior to conducting the experiment, we ran tests to assess the potential for side bias in our experimental arena. The test followed the procedures described above but both light treatments were off on both sides of the arena. Each of the 23 individuals were exposed to the test arena on three different occasions. We randomized the order of

exposure across individuals. If an animal did not make a side choice within 10 min, the test trial was excluded from the analyses.

Using an intercept-only generalized linear mixed model (*i.e.,* no independent factors), with the identity of the individual as a random factor and whether individuals chose the right (1) or left (0) side of the arena as the dependent factor, we found that there was no significant difference in the probability of going right (intercept estimate $-0.36 \pm 0.26$, z $= -1.41$, $P = 0.158$), suggesting there was no side bias in our arena. This provided support that our experimental arena did not have a side bias. The code for the analysis can be found at https://osf.io/g9am5/?view_only=a5c667733e044a8090a724cce413b30b.

## Behavioral analysis

We analyzed the behavior of the focal individual frame by frame with the Avidemux video player (*Avidemux-Main Page, 2022*). From the videos, we estimated latency to respond to the treatments, head movement rate, and body movement rate before the choice took place, and corroborated the side of the arena the animals chose. Quantifying changes in latency, head and body movement rate has implications for better understanding animal decision making in the process of initiating and enacting avoidance responses (*Bulbert, Page & Bernal, 2015*; *Card & Dickinson, 2008*; *Tomsic & Theobald, 2023*).

Latency to respond in seconds was defined as the total duration in seconds from the time the goose entered into the arena (*i.e.,* the beginning of the trial) to the time it made a choice (*i.e.,* the end of the trial as captured from the perspective of the east camera). We defined the beginning of the trial as the first frame where the gate of the release enclosure elevated to 90° relative to the door of the release enclosure, providing the goose with unobstructed visual access to the experimental arena. As noted, we defined the end of the trial as the first frame where the beak of the goose passed the beginning of the partition and crossed into either the left or right side of section 2 in the arena (Fig. 1A).

We measured the number of distinct changes in both head and body orientation during each trial before the animal made a choice from the perspective of the west camera (Fig. 1B). However, the positioning of the west camera provided a relatively limited viewing angle, and it did not fully capture the exact moment of the beginning of the trail (as previously defined). Despite this shortcoming, we chose not to use the east camera because the view of the animal was partially blocked when the animal was in the center of section 1 and did not have enough resolution to measure subtle changes in head and body movement as the animal moved through the experimental arena. Therefore, we measured head and body orientation changes after the individual first appeared on the west camera instead of the very beginning of the trial. We ultimately estimated head movement rate (number of events per second) and body movement rate (number of events per second) as the frequency of distinct movements divided by the time the animal was visible to the west camera.

We defined a change in head orientation as any distinct change in yaw, pitch, or roll relative to the previous head orientation of the animal (Fig. 1B) (*Dawkins, 2002*; *Fernández-Juricic & Kowalski, 2011*; *Moore et al., 2017*). For example, if the beak was pointed directly at the partition, but the goose began to turn its head and the beak stopped at 90° in the yaw axis we considered that movement to be one change in head orientation (Fig. 1B)

(*Fernández-Juricic & Kowalski, 2011*; *Moore et al., 2017*). In the case where the animal continued to move its body forward through the arena in a single direction but did not change the orientation of its head, we considered that to be no change in head orientation.

We defined a change in body orientation as any distinct change in the rotation of the body that would result in a deviation from the animal's prior trajectory. For example, if the body was directed at the partition, but the animal turned moving its feet or rotating its torso 90° to the right, stopped, and faced the south wall of the arena, we considered this movement as one change in body orientation (Fig. 1B). Minor changes such as a ruffling of tail feathers or opening of wings were not counted as changes in body orientation. If the animal continued to move in a single continuous trajectory forward, we considered that to be no change in body orientation.

Attraction or avoidance was measured based on the location of the animal within the arena upon the end of the trial (*i.e.,* the moment the animal crossed the decision threshold established by the partition). We recorded the moment the animal crossed the decision threshold ending the trial into the side of the arena with a light stimuli as an attraction response whereas when animals went away and crossed the decision threshold of the arena as an avoidance response. Again, we used this directional choice as a proxy to establish attraction or avoidance responses to the light. We coded the choice to move toward or away from the light stimulus as 0 and 1 respectively.

## Statistical analysis

We conducted statistical analyses and created figures representing our data in R version 4.2.1 (*R Core Team, 2022*). All code and data for this study are available for download at the Open Science Framework (https://osf.io/g9am5/?view_only=a5c667733e044a8090a724cce413b30b).

We began by assessing potential multicollinearity issues within the confounding factors we measured (time of day, ambient light intensity on the side with the light on and with the light off, temperature). Ambient light intensity has been shown in previous studies to affect the perception of LED lights by birds (*Blackwell et al., 2009b*; *Blackwell et al., 2012*; *Kristensen et al., 2007*; *Rebke et al., 2019*). We found a positive association between ambient light intensity on the side of the arena with the light on and the side with the light off (Pearson's product moment correlation; $r = 0.83$, $P < 0.001$), thus we decided to run a principal component analysis (PCA) including these two variables (*i.e.,* ambient light intensity on the side with the light on and ambient intensity on the side with the light off) to summarize their effects. The PCA identified a single factor with an Eigenvalue higher than 1 (PCA1, 1.83), which explained 91.4% of the variation. Both ambient light intensity on the side of the arena with the light "on" ($r = 0.96$, $P < 0.001$) and ambient light intensity on the side with the light "off" ($r = 0.96$, $P < 0.001$) were positively correlated with PCA1. Therefore, higher values of PCA1 (hereafter, PCA ambient light intensity) were indicative of higher ambient light intensity on both sides of the arena. Both temperature ($r = 0.18$, $P = 0.03$) and time of the day ($r = -0.46$, $P < 0.001$) were significantly correlated with PCA ambient light intensity. To reduce the chances of collinearity in our models, we decided to exclude temperature and time of the day from subsequent analyses. Furthermore, we

use the coordinates of PCA 1 as the independent variable for ambient light intensity in all subsequent analyses.

Because of our randomization of light position in the first four trials and, subsequently, selecting the opposite side of the arena for the light-on position in the final four trials, there was potential for an association between light on position (right, left) and the other categorical factors included in our design (light color, light pulsing frequency, trial order). We ran a generalized linear model with light-on position (right, 1; left, 0) as the dependent variable and three independent categorical variables: light color, light pulsing frequency, trial order, and all their potential interactions. We found a significant two-way interaction between color and trial order ($X^2{}_7 = 62.69$, $P < 0.001$) and a three-way interaction among color, pulsing frequency, and trial order ($X^2{}_7 = 25.54$, $P < 0.001$). Because of this association, we chose to remove light on position in the arena from all subsequent models.

As a result of removing temperature, relative humidity, and light position, our base model included four independent factors: three categorical (light color, light pulsing frequency, trial order) and one continuous (the PCA coordinates of ambient light intensity). We included trial order in our base model as well to account for changes in animal behavior upon repeated exposures to the experimental arena (*Blumstein, 2016*; *Rankin et al., 2009*). Additionally, because all individuals were repeatedly exposed to all the same treatments, we needed to account for the effect of individual differences in behavioral responses and used a mixed modelling approach for our statistical analysis.

We used general and generalized linear mixed models, run with the R package *afex* (*Singmann & Kellen, 2019*), to analyze four dependent variables: latency to respond to the lights (s), head movement rate (events per second), body movement rate (events per second), and the probability of avoidance (*i.e.,* higher probabilities indicate higher chances of avoiding the light treatment). The statistical test to determine significance was either a Kenward-Rogers approximation for the general linear mixed models (*i.e.,* latency, head movement rate, and body movement rate) or the log-likelihood ratio test for the generalized linear mixed model (*i.e.,* probability of avoidance). We checked for the homogeneity of variance and normality of the error assumptions for latency to respond to the light stimulus, head movement rate, and body movement rate.

Latency to respond to the light stimulus model did not meet the normality of error and homogeneity of variance assumption. A log-transformation slightly improved the model fit to the assumptions; however, there is a distinct possibility that transformation of the data could ameliorate interaction effects (*Schielzeth et al., 2020*). Therefore, given the robustness of general linear models (*Schielzeth et al., 2020*), we present the untransformed data to facilitate the interpretation of the results, particularly relative to interaction effects (*Belzak & Bauer, 2019*). In all models, we included individual bird identity as a random factor. The random structure of our mixed model consisted of only random intercepts (*i.e.,* (1|bird id)). Unfortunately, we were unable to increase the complexity of the random structure by adding random slopes (*i.e.,* to examine variation in slope across groups) due to lack of model convergence, which was likely caused by the relatively limited sample size across treatments. Our sample size was primarily limited by the size of the housing enclosure needed to maintain high standards of animal husbandry. Specifically we wanted to maintain

2.7 square meters per a single goose (*i.e.,* 30 square feet) to minimize aggression between individuals (*Gleaves, 1984*). Twenty-three geese were all that we could accommodate at that time (*i.e.,* 65 square meters divided by 2.8 m per goose equals a maximum of 23 individuals).

It is important to note that while all our models converged (Table S3), the models for only the probability of avoidance produced singular solutions. Unfortunately, it was not possible to further reduce the complexity of the random structure, as suggested by *Singmann & Kellen (2019)* because we were already using the simplest random structure (*i.e.,* there were no higher-order random effects that could be removed). Following *Singmann & Kellen (2019)*, we proceeded because our analysis was focused on the fixed effects (*i.e.,* the differences in light treatments), and the random structure (*i.e.,* individual ID) was needed to properly account for the lack of independence between trials (*i.e.,* repeated measures).

Due to sample size, we were limited in our ability to test for all possible interaction effects. For instance, by including single effects and all possible interaction effects, we would have 15 independent factors in our model, running the risk of over parameterizing. Consequently, we decided to evaluate each two-way interactions in a step-wise method divided in four steps. First, we included all four single, independent factors as well as the interaction between light color and light pulsing frequency as it reflected both independent variables of interest in our experimental design. If the interaction was not significant, we removed it prior to the next step. Second, we included the four single independent factors, our base model, (as well as the interaction if significant from step 1) and included the interactions between light color and trial order and the interaction between light pulsing frequency and trial order. Non-significant interactions were removed prior to the next step. Third, we included the four single independent factors (as well as the significant interaction(s) from steps 1 and 2) and the 2-way interactions between light color and PCA ambient light intensity and the interaction between light pulsing frequency and PCA ambient light intensity. Fourth, we ran our final model keeping the single independent factors but removing all the non-significant interactions from the previous steps. The final model we report for each dependent variable is the base model (light color, light pulsing frequency, trial order, and the PCA coordinates of ambient light intensity) with all the significant interactions found in the stepwise process described in this paragraph.

We used the R package *emmeans* (*Lenth et al., 2019*) to estimate the least square means and SEs for different treatments. We used the function afex_plot from the R package *afex* (*Singmann et al., 2015*) to plot our results. We reported marginal $R^2$, conditional $R^2$, and the differences in between individual variation, what some studies refer to as repeatability (*Dingemanse & Dochtermann, 2013*; *O'Dea, Noble & Nakagawa, 2021*; *Nakagawa & Schielzeth, 2010*; *Stoffel, Nakagawa & Schielzeth, 2017*; *Wolak, Fairbairn & Paulsen, 2012*). The marginal $R^2$ is a measure of effect size which explains the amount of variance in the dependent variable explained by only the fixed factors in a mixed model (*Nakagawa & Schielzeth, 2013*). In contrast, the conditional $R^2$ is a measure of effect size which explains the amount of variance in the dependent variable explained by both the fixed and random factors in a mixed model (*Nakagawa & Schielzeth, 2013*). We estimated differences in between-individual variation from the variance associated

with the random effects (*i.e.,* individual ID) divided by the sum of all the variance observed. The total variance includes the variance of the random effects and the variance of the residuals controlled for with the fixed effects. We used the following equation to estimate repeatability or the variation within the data accounted for by between-individual differences $\frac{V_{individual}}{V_{individual}+V_{residual}}$ (*Nakagawa & Schielzeth, 2010*; *Dingemanse & Dochtermann, 2013*; *O'Dea, Noble & Nakagawa, 2021*). Each value was multiplied by 100 to convert the proportions to percentages. Following (*Bell, Hankison & Laskowski, 2009*; *Wolak, Fairbairn & Paulsen, 2012*; *Baker et al., 2018*), we categorized the values for between individual differences as either low if the value was less or equal to 20%, moderate if the value was greater than 20% or equal to or less than 40%, all other values greater than 40% were considered high. In essence, smaller values generally mean that individuals tended to have similar responses to the treatment where larger values suggest that individuals tended to have different responses suggesting that the responses are specific to the individual.

## RESULTS

### Latency to respond

The final model for latency to respond (s) included four variables: light color, light pulsing frequency, trial order, and PCA ambient light intensity (with higher values representing higher light intensity on both sides of the arena). No interaction effects were included in the final model per our selection procedure (see Methods). Latency to respond was 13.4 s faster for the red light (28.6 ± 11.4 s) compared to the blue light treatment (42.0 ± 11.4 s), but the difference between the treatments was not significant (Table 1). Additionally, latency to respond was faster for the pulsing light (23.3 ± 11.5 s) than to the steady light (47.3 ± 11.5 s) but again the difference was not significant (Table 1). Latency among trials also did not vary significantly (trial 1, 49.1 ± 19.5 s; trial 2, 29.9 ± 19.2 s; trial 3, 21.1 ± 20.5 s; trial 4, 38.2 ± 19.0 s; trial 5, 60.0 ± 19.3 s; trial 6, 42.1 ± 20.6 s; trial 7, 23.4 ± 19.2 s; trial 8, 18.8.1 ± 20.3 s). Lastly, latency had a positive association with PCA ambient light intensity (*i.e.,* geese tended to move slower in brighter conditions) (coefficient estimate 4.8 ± 6.0 s), but it was not significant (Table 1).

The marginal $R^2$, which only considers the fixed effects, explained 5.3% of the variation in latency, *versus* the conditional $R^2$ which considers both fixed and random effects, explained 19% of the variation. Focusing on just the random effects, we estimated between-individual variation to account for 14.4% (CI [8.8%–21.5%]) of the variation in latency to respond, a low value (Fig. S2A).

### Head movement rate

The final model for head movement rate (events per second) included four independent variables: light color, light pulsing frequency, trial order, and PCA ambient light intensity, without interaction effects (Table 1). Head movement rate was 11% higher with the red light (1.2 ± 0.01 events per second) compared to the blue light (1.06 ± 0.01 events per second), but the difference was not significant (Table 1). The difference in head movement rate between the pulsing light (1.13 ± 0.01 events per second) compared to the steady light (1.11 ± 0.01 events per second) was not significant (Table 1). Head movement rate did

**Table 1 The effects of color, pulsing frequency, trial order, and ambient light condition on latency, head movement rate, body movement rate, and the probability of avoidance.** Results from both general and generalized linear mixed models (significant values are bolded).

| General Linear Mixed Model Results | F | d.f | P |
|---|---|---|---|
| **Latency (s)** | | | |
| *Color* | 1.15 | 1, 123.06 | 0.286 |
| *Frequency* | 3.55 | 1, 123.06 | 0.062 |
| *Trial Order* | 0.69 | 7, 123.53 | 0.682 |
| *Light intensity PCA* | 0.63 | 1, 131.74 | 0.428 |
| **Head movement rate (events per second)** | | | |
| *Color* | 1.58 | 1, 123.06 | 0.211 |
| *Frequency* | 0.06 | 1, 123.06 | 0.803 |
| *Trial order* | 0.97 | 7, 123.55 | 0.457 |
| *Light intensity PCA* | 0.23 | 1, 131.13 | 0.632 |
| **Body movement rate (events per second)** | | | |
| *Color* | 0.85 | 1, 123.05 | 0.358 |
| *Frequency* | 0.82 | 1, 123.05 | 0.367 |
| *Trial order* | 2.81 | 7, 123.46 | **<0.009 \*\*** |
| *Light intensity PCA* | 0.01 | 1, 130.77 | 0.928 |
| *Generalized Linear Mixed Model Results* | $X^2$ | d.f | P |
| **Probability of light avoidance** | | | |
| *Color* | 6.35 | 1, 19 | **0.011\*** |
| *Frequency* | 0.41 | 1, 19 | 0.521 |
| *Trial order* | 15.85 | 7, 13 | **0.026\*** |
| *Light intensity PCA* | 0.12 | 1, 19 | 0.731 |
| *Color & Trial Order Interaction* | 29.07 | 7, 13 | **<0.001\*\*\*** |
| *Frequency & Light Intensity PCA Interaction* | 9.26 | 1, 19 | **<0.003\*\*** |

not vary significantly among trial exposures (trial 1, 0.93 ± 0.14; trial 2, 0.96 ± 0.14; trial 3, 1.09 ± 0.15; trial 4, 1.25 ± 0.14; trial 5, 1.07 ± 0.14; trial 6, 1.09 ± 0.15; trial 7, 1.25 ± 0.14; trial 8, 1.30 ± 0.15 events per second). Lastly, head movement rate had a weak, non-significant, negative association with PCA ambient light intensity (coefficient estimate −0.02 ± 0.04 events per second; Table 1).

The marginal $R^2$ explained 4.5% of the variation in head movement rate; whereas the conditional $R^2$ explained 17.3% of the variation. Focusing on just the random effects, we estimated between-individual variation to account for 13.3% (CI [7.7%–19.8%]) of the variation in head movement rate, again a low value (Fig. S2B).

## Body movement rate

The final model for body movement rate (events per second) included four independent variables: light color, light pulsing frequency, trial order, and PCA ambient light intensity, without interaction effects (Table 1). Individuals increased their body movement rate by 14% in response to the red light (0.48 ± 0.06 events per second) compared to the blue light (0.42 ± 0.05 events per second), but the difference was not significant (Table 1). Body movement increased by 14% in response to the pulsing light (0.48 ± 0.06 events

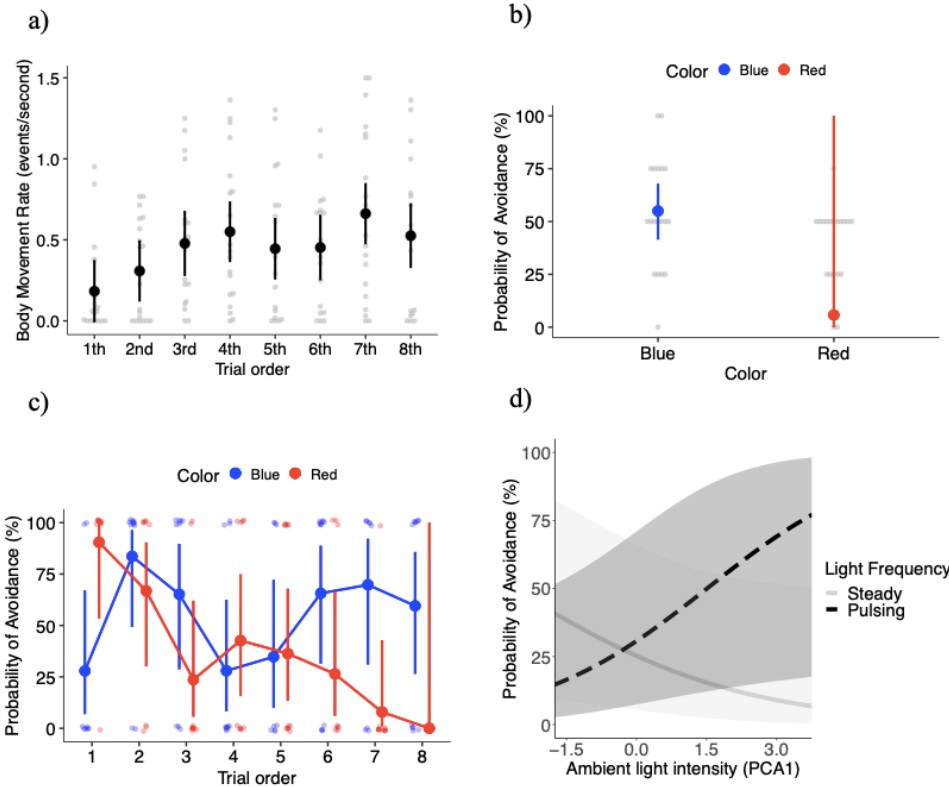

**Figure 2 Key results of the behavioral experiment.** (A) Mean ± SE body movement rate (events per second) relative to trial order. Gray dots represent the raw data. Probability of avoiding lights (mean estimates ± SE) relative to: (B) light color (blue and red lights), (C) the interaction between light color and trial order, and (D) the interaction between light frequency and ambient light intensity (represented by the first principal component analysis factor).

per second) compared to the steady light (0.42 ± 0.05 events per second), but without significant effects (Table 1). However, body movement rate varied significantly with trial order (Table 1), with a trend towards more body movements with increasing exposures to the treatment conditions (Fig. 2A). Lastly, body movement rate had a weak negative association with PCA ambient light intensity (coefficient estimate −0.002 ± 0.02 events per second) that was not significant (Table 1).

The marginal $R^2$, explained 10.8% of the variation in body movement rate; whereas the conditional $R^2$, explained 26.4% of the variation. Focusing on just the random effects, we estimated between-individual variation to account for 17.5% (CI [10.5%–25.1%]) of the variation in body movement rate, again slightly higher but still a low value (Fig. S2C).

## Probability of light avoidance

The final model for the probability of avoidance included: light color, light pulsing frequency, trial order, PCA ambient light intensity, the interaction between color and trial order, and the interaction between pulsing frequency and PCA ambient light intensity. The probability of avoidance was significantly higher, with a 49% increase in the probability of

avoidance in response to the blue light compared to the red light ($X^2_1 = 6.35$, $P = 0.012$; Fig. 2B). Figure 2B shows a large standard error for the red light, which is likely the result of the high level of variation in the response to the red light over the course of the experiment (see below). The probability of avoidance was 4.6% higher with the light pulsing ($0.24 \pm 15.1$) than the light steady ($0.19 \pm 13.0$), but the differences were not significant ($X^2_1 = 0.41$, $P = 0.522$). The probability of avoidance varied significantly with trial order ($X^2_7 = 15.85$, $P = 0.026$).

While the effects of light color were significant, its effects depended on trial order, as the interaction between light color and trial order was significant ($X^2_7 = 29.07$, $P = 0.00014$; Fig. 2C). Overall, there was a trend towards a high probability of avoidance to the red light at the beginning of the experiment (close to 0.90), but then a steady decrease as the experiment progressed with probability of avoidance close to 0 at the very end of the experiment (Fig. 2C). Because the probability of attraction can be estimated from 1 - probability of avoidance, another interpretation is that Canada geese upon repeated exposures to the red light developed an attraction to it (Fig. 2C). On the other hand, the probability of avoidance to the blue light oscillated to a larger degree over the course of the experiment. During the first trial geese tended to go towards the blue light, however trials 2 through 8 demonstrated a U-shaped pattern where the probability of avoidance was higher in trials 2 and 3, geese tended to go towards the light in trials 4 and 5. Finally, the average probability of avoidance was 65% in the last three exposures. More specifically, the difference in the probability of avoidance between blue and red lights was significantly different in trial 1 (z ratio = $-2.34$, $P = 0.019$), with geese showing higher probability of avoidance for the red relative to the blue light (Fig. 2C). In trial 7 (z ratio = 2.29, $P = 0.022$), geese showed higher probability of avoidance for the blue relative to the red light (Fig. 2C). Again, the large standard errors are likely the result of the variability in responses within specific treatment combinations.

PCA ambient light intensity did not significantly affect the probability of avoidance with a weak positive association (coefficient estimate $0.07 \pm 0.19$) ($X^2_1 = 0.12$, $P = 0.732$). However, the interaction between PCA ambient light intensity and light pulsing frequency was significant ($X^2_1 = 9.26$, $P = 0.002$). Under brighter ambient light conditions individuals were more likely to avoid the pulsing light, but we found the opposite trend relative to steady lights (*i.e.,* lower probabilities of avoidance with brighter ambient light conditions) (Fig. 2D). We estimated between-individual variation to account for only 2.7% of the variation in the probability of avoidance, an extremely low value (Fig. S2D).

## DISCUSSION

Our results suggest that Canada geese responded differently to high visual contrast lights of different colors and pulsing frequencies relative to the number of exposures and ambient light conditions. Specifically, Canada geese had an overall higher probability of avoidance in response to blue light compared to red light. However, the probability of avoidance changed substantially with repeated exposure to the light stimuli. Canada geese went from avoidance to attraction to the red light over the course of the experiment. The response to

the blue light generally followed a U shape relationship (avoidance, attraction, avoidance) with increasing number of exposures. If the threshold difference between avoidance and attraction is either greater or less than a 50% probability of avoidance, respectively, individuals were attracted to the red light 75% of the time out of eight trials where in contrast they avoided the blue light 63% of the time (attracted only 37% of the time) out of eight trials. This trend was particularly pronounced towards the end of the experiment (trials 6–8; Fig. 2C) where the mean probability of attraction to the red light was 11% ± 14, and the mean probability of avoidance of the blue light was 65% ± 5. Additionally, we found that the probability of avoidance increased in response to a pulsing light (of either color) in brighter ambient light conditions, whereas avoidance of a steady light (of either color) increased in dimmer ambient light conditions. Lastly, individuals regardless of the light treatment initially increased body movement rate, which then plateaued across the subsequent trials.

Light stimuli with different peak wavelengths led to different probabilities of avoidance despite both exceeding the threshold chromatic contrast required to detect an object based on modelling of the Canada goose visual system (*Vorobyev & Osorio, 1998*). The threshold to discriminate a visual stimulus from the background is suggested to be between 1–4 JNDs (*Vorobyev et al., 2001*; *Vorobyev & Osorio, 1998*). However, the chromatic contrast of the LED lights used in this study far exceeded these thresholds (*i.e.,* the blue light was 25 and the red light was 45 JND, Supplemental Information 1). In principle, this finding suggests that greater retinal stimulation, which in theory connotes a more conspicuous stimulus, could lead to a greater degree of behavioral responses (*Endler et al., 2022*; *Fleishman et al., 2016*; *Santiago et al., 2020*). Empirical evidence in lizards (*Anolis sagrei*) suggests that chromatic contrast has a linear relationship with the probability of detection (*i.e.,* eye fixations) (*Fleishman et al., 2016*). However, in coral reef fish (*Rhinecanthus aculeatus*) JND values <10 had a linear relationship with detection, measured *via* pecking behavior (*Santiago et al., 2020*). However, at larger JND values (≥10) the association between chromatic contrast and detection-related behavior plateaus (*Santiago et al., 2020*). The fact that we found different types of behavioral responses for light stimuli with different chromatic contrast far above 10 JND suggests a lack of understanding in how retinal stimulation above detection thresholds is associated with cognition/perception and the corresponding behavioral response.

A key finding was that the probability of avoidance changed substantially upon repeated exposures and those responses changed depending on the wavelength of the light stimuli. In the case of the red light, geese went from avoiding it in the first trial to being attracted to it by the last trial. This trend mimics the mere exposure effect (*Zajonc, 1968*), whereby individuals tend to be more cautious and avoid a given treatment stimulus upon first exposure, but after subsequent exposures, individuals increase their familiarity to it eventually developing an attraction response (*Fang, Singh & Ahluwalia, 2007*; *Montoya et al., 2017*; *Zajonc, 1968*). Evidence of the mere exposure effect has also been found in chickens (*Gallus domesticus*; *Franchina, 1991*) and turkeys (*Meleagris gallopavo domesticus*; *Sherwin, 1998*) as well as other non-human animals (rats *Rattus norvegius*, mice *Mus musculus*, Japanese macaque *Macaca fuscata,* cats *Felius catus; Hill, 1978*; *Bradshaw, 1986*), in the context of attraction

or avoidance related behaviors. The attraction to the red light after repeated exposures cannot be characterized as habituation, primarily because geese kept responding to the light eventually developing an attraction (*i.e.,* not a neutral response).

In the case of the blue light, the response of geese over the course of the experiment appeared to follow the mere exposure effect followed by a satiation effect (*Bornstein, Kale & Cornell, 1990*). Individuals in our study went from initial, low avoidance, but increased in avoidance by trial 2. Later, the birds showed attraction (*i.e.,* mere exposure effect), but then became overexposed to the stimuli and ultimately developed an avoidance response again (*Bornstein, Kale & Cornell, 1990*; *Montoya et al., 2017*). Disregarding the first trial where geese were attracted to the blue light (probability of avoidance was 27%), trials 2 through 8 can be generally characterized as a U-shaped pattern, whereas trials 2 and 3 had a relatively higher probability of avoidance. Avoidance decreased for trials 4 and 5, and then finally increased and remained higher in trials 6, 7, and 8 (Fig. 2C). The decrease in the probability of avoidance suggests that the birds became more familiar with the treatment upon repeated exposures and therefore were more attracted to the blue light in trials 4 and 5. However, instead of developing an attraction (*i.e.,* a continued decrease in the probability of avoidance like the red light), the probability of avoidance increased and remained comparatively stable at 65% during the last three trials, suggesting the birds had satiated to the blue light (*i.e.,* weak avoidance).

One proposed explanation for the mere exposure effect is the processing fluency model that argues that the transition from neophobic avoidance to attraction occurs because stimuli become cognitively easier to process with repeated exposures (*Lodge & Cottrell, 2010*; *Montoya et al., 2017*; *Reber, Schwarz & Winkielman, 2004*; *Wänke & Hansen, 2015*; *Winkielman et al., 2003*). Humans and non-human animals are more likely to detect and react faster to high contrast stimuli (*Blough, 2000*; *Blough, 2002*; *Kurylo et al., 2015*) suggesting that higher contrast stimuli are easier to process (*Leynes & Addante, 2016*; *Reber, Schwarz & Winkielman, 2004*). Our red light had a higher chromatic contrast (45 JND) compared to the blue light (25 JND); a difference that could have made the red light easier to process visually and cognitively, possibly leading to the development of an attraction response. In comparison, the relatively lower chromatic contrast of the blue light might have resulted in a higher cognitive load to process in relative terms. Animals have a limit to the amount of information they can process per unit time (*Dukas, 2004*). We argue that the potentially higher cognitive processing costs led to an increase in avoidance responses to the blue light compared to the red light.

Other studies have reported observational evidence of both attraction and avoidance responses to red and blue lights amongst various bird species (Table S4). Using the systematic map established by *Adams et al. (2021)* and non-systematically searching for other studies, we identified 13 different papers that entailed a total of 26 different experiments/studies (Table S4). Fourteen studies investigated behavioral responses to red lights: 50% found evidence to suggest that birds were attracted to red light, whereas the other 50% suggest that birds tended to avoid red light. Twelve studies investigated behavioral responses to blue lights: 25% found evidence to suggest that birds were attracted to blue light, whereas the other 75% suggest that birds tended to avoid the blue light. This

collection of studies suggests that an avoidance response to blue light is more common than red light, a trend our study supports. However, these results should be taken with extreme caution as (1) our search of the literature was not systematic, and (2) only one other study including this one manipulated the choices and made them mutually exclusive (see *Goller et al., 2018*). Many of the 26 studies were not able to control for confounding factors such as identity of individuals, local abundance of the species, *etc.* In addition, the 26 studies cover a wide range of scenarios from different species, different times of day, different environments, and different light types. Nevertheless, these findings, in combination with our own, raise the question as to what ultimate factors make a species, in our case the Canada goose, avoid or approach certain wavelengths. At this early stage in our understanding of avian responses to novel light stimuli, we are not in a position to make specific conclusions.

Canada geese had a higher probability of avoiding a flashing light (irrespective of color) under higher ambient light intensity, but a higher probability of avoiding a steady light under lower ambient light intensity. This result follows the trends of *Blackwell et al. (2012)*, who measured Canada geese alert distance to an approaching aircraft with a 2 Hz pulsing light and found that geese alerted sooner to a pulsing light under brighter ambient light conditions. The result is also similar to *Doppler et al. (2015)* where Brown-headed cowbirds reacted faster to an aircraft with a pulsing light stimulus. In contrast *Blackwell et al., (2009b)*, found that Brown-headed cowbirds and Mourning doves (*Zenaida macroura*) reacted sooner to an approaching vehicle with a light stimulus pulsing at 2 Hz and 16 Hz in dim light, whereas in brighter ambient conditions they responded sooner to a steady light. Importantly, *Blackwell et al. (2009b)*, *Blackwell et al. (2012)*, and *Doppler et al. (2015)* measured alert responses, whereas our data pertain to choice responses. Still, the overall implication is that the response to pulsing light varies with ambient condition and species, but it appears that initially Canada geese find a pulsing light more conspicuous in brighter ambient conditions. Perhaps a pulsing light stimulus appears more conspicuous to geese when the signal is spaced out temporally (*i.e.,* pulsing) in contrast to when the light is constant and ambient light in the daytime is abundant.

Differences in individual experience are sometimes a factor accounting for between-individual differences (*Dingemanse & Wolf, 2013*; *Dukas, 2017*; *Sih, Sinn & Patricelli, 2019*). The between-individual variation values for all four dependent variables (latency, 14.4%; head movement rate, 13.3%; body movement rate, 17.5%; probability of avoidance 2.7%) were considered low. Variation in between-individual differences for the probability of avoidance was 2.7% suggesting that between-individual variation in Canada geese likely has a limited effect on the response to light stimuli. The extremely low value for the percentage of the variation in the probability of avoidance attributable to between-individual variation suggest that geese with the same experience (*i.e.,* number of exposures to light stimuli) would tend to have similar avoidance responses. These findings should be taken cautiously primarily because it is more difficult to resolve between-individual variation in responses for binary variables (*i.e.,* each trial the animal chose between one of two choices) compared to continuous variables (*Nakagawa & Schielzeth, 2010*). Furthermore, this metric of between-individual variation is imperfect as it is difficult to discern whether a relatively lower

value is the result of large within-individual variation in response to the treatment or the result of little variation between individuals in response to the treatment (*Dochtermann & Royauté, 2019*). Lastly, because the random structure of our models was limited to random intercepts only our estimates for between-individual variation were limited and do not account for how different individuals might have altered their responses to different treatments (*i.e.,* random slopes). However, the combination of low values for the effect of between-individual variation for each dependent variable suggests that the effects of lights on goose behavior are generalizable in that we would expect that responses tend to converge. Further research is needed to determine the effect of low between individual variation in response to light stimuli has on the probability of collisions.

Geese also increased body movement rate upon the first three trials where body movement rate then plateaued and remained relatively consistent over the remaining five trials. The increase in body movement rate was significant but might be a residual artifact of the biomechanics of waterfowl bi-pedal locomotion within terrestrial environments. Waterfowl terrestrial locomotion is often characterized by waddling or horizontal shifts as the trunk moves over the foot when walking forward to support the animals center of gravity (*Daley & Birn-Jeffery, 2018*; *Provini et al., 2012*). Evidence suggest that an increase in Mallard duck (*Anas platyrhynchos*) movement speed is accompanied by an increase in movement amplitude and stride length (*Abourachid, 2000*; *Provini et al., 2012*). While not significant there was a small trend towards a decrease in latency to respond with an increase in trial order (Fig. S3A). A shorter latency to respond in general requires individuals to move faster in a continuous direct motion past the partition to either the left or right side of the arena. Geese adopting a slightly faster movement speed likely had greater amplitude in the horizonal shifts of the torso. A relatively larger variation in horizontal shifts with faster walking speeds might have led to more directional variation as the animal moved forward in the arena. One potential explanation is that when the animals had less experience in the arena, they were more cautious and moved slower (*i.e.,* neophobia). Subsequent exposures (trials 1–3) to the arena resulted in a decrease in neophobia which might have led to an increase in walking speed which was accompanied by an increase in body orientation changes. The experimental paradigm forced the animal to make a left or right directional choice. It is possible individual geese needed to make more distinct shifts in body orientation to maintain the correct directional position as they moved faster towards the side of the arena they selected.

While not significant, it is important to note that the differences in latency between light treatments showed effect sizes that may be relevant in the context of high-speed vehicles. Animals responding to a high-speed vehicle at close distances or when the vehicle is a few seconds away is critical to determine whether a collision may occur or not (*DeVault et al., 2015*; *Brieger et al., 2022*). The difference in mean latency between the red and the blue light treatments was about 13 s, whereas the difference between the pulsing and steady light treatments was about 24 s. This temporal difference could have implications for the probability of collision. For example, if a goose begins to escape from an approaching aircraft at either 13.4 or 24 s prior to the vehicles arrival and the aircraft is travelling at 66.66 m/s (*i.e.,* a standard take-off and landing speed) that translates to the animal being aware

of or beginning to escape at a distance of either 893 m or 1600 m away from the aircraft. In both of those circumstances a difference of even a few seconds results in substantial change in the reaction distance, reducing the chances of a collision. Importantly, our experimental design purposely lacks factors that influence how a bird responds to an approaching vehicle (*e.g.*, size, looming effect, aspects of visual flow), as our objectives were specific to assessing just a neutral light stimuli. That said, the potential implication of these differences in latency to respond between different light treatments warrants further investigation.

## CONCLUSIONS

Our study has implications not only for onboard lighting systems aimed at deterring bird strikes but also for reducing collisions between birds and other anthropogenic structures (*e.g.*, buildings, wind turbines, *etc.*). First, our results provide additional evidence that lights of high chromatic contrast peaking in the range of 464 nm to 483 nm can elicit avoidance responses in bird species with different types of visual systems (Canada goose, violet-sensitive species, our study; Brown-headed cowbird, ultra-violet sensitive species, (*Goller et al., 2018*). Exploring the behavioral responses to LED lights peaking around the blue portion of the spectrum in a more systematic way appears the next step to potentially enhance avian avoidance responses. Second, red LED lights have the unwanted potential to develop strong attraction responses, at least in Canada geese, based on the number of times individuals are exposed to it. Because it is challenging to estimate the degree of experience with LED lights for different individuals within a bird population, if we are to apply the precautionary principle, we suggest avoiding this portion of the spectrum as it has the potential to increase the frequency of bird strikes due to attraction effects. Third, given that the avoidance effects of light pulsing frequency are a function of ambient light, we suggest that light deterrent systems should incorporate systems that automatically modify the pulsing of the light depending on ambient light intensity to enhance behavioral responses. Establishing the thresholds of light intensity that switch the behavioral responses to light should be considered before deterrence implementation.

## ACKNOWLEDGEMENTS

We are deeply grateful for Benny Goller and Justin Vickery's aid in constructing the aviary which housed the animals and the experimental arena. Additionally, we would like to thank Morgan Chaney, Benny Goller, Carlay Teed, Deona Harris and Becca Trapp for their help in conducting trials. Lastly, I would like to thank one of my most important collaborators, Becca Lunn, who not only assisted in conducting experimental trials but also provided additional logistical support which made this experiment possible.

### Funding

Our work was funded *via* the Cooperative Agreement with the U.S. Department of Agriculture, Animal and Plant Health Inspection Service, Wildlife Services (WS), National

Wildlife Research Center (FAIN: AP22WSNWRC00C006), and based on funding received by WS *via* the Interagency Agreement with the U.S. Federal Aviation Administration (FAA Interagency No. 692M15-19-T-00017). Findings reported herein do not necessarily reflect the policy of the FAA. The development of the light stimulus at Rensselaer's Lighting Research Center was funded by the Federal Aviation Administration (FAA) under Cooperative Agreement Number 692M151940010 "Lighting and Visual Guidance Research for Airport Applications." The funders had no role in study design, data collection and analysis, decision to publish, or preparation of the manuscript.

## Grant Disclosures

The following grant information was disclosed by the authors:
Cooperative Agreement with the U.S. Department of Agriculture, Animal and Plant Health Inspection Service, Wildlife Services (WS), National Wildlife Research Center: FAIN: AP22WSNWRC00C006.
Interagency Agreement with the U.S. Federal Aviation Administration: FAA Interagency No. 692M15-19-T-00017.
The Federal Aviation Administration (FAA) under Cooperative Agreement Number 692M151940010 "Lighting and Visual Guidance Research for Airport Applications.".

## Competing Interests

The authors declare there are no competing interests.

## Author Contributions

- Ryan Lunn conceived and designed the experiments, performed the experiments, analyzed the data, prepared figures and/or tables, authored or reviewed drafts of the article, and approved the final draft.
- Patrice E. Baumhardt analyzed the data, authored or reviewed drafts of the article, contributed the chromatic contrast calculations, and approved the final draft.
- Bradley F. Blackwell conceived and designed the experiments, authored or reviewed drafts of the article, secured funding, and approved the final draft.
- Jean Paul Freyssinier conceived and designed the experiments, authored or reviewed drafts of the article, built the light stimulus, and approved the final draft.
- Esteban Fernández-Juricic conceived and designed the experiments, analyzed the data, authored or reviewed drafts of the article, secured funding, and approved the final draft.

## Animal Ethics

The following information was supplied relating to ethical approvals (i.e., approving body and any reference numbers):
Institutional Animal Care and Use Committee at Purdue University.

## Data Availability

The raw data and the code for all analyses used in this experiment including the generation of figures used for this experiment are available in the Supplementary Files.
The code and data is also available at GitHub and Open Science Framework:

- https://github.com/ryanlunn/CanadaGooseLightAvoidance.git

- Lunn, Ryan, and Esteban Fernandez-Juricic. 2023. "Light Wavelength and Pulsing Frequency Affect Avoidance Responses of Canada Geese." OSF. September 20. osf.io/g9am5.

## Supplemental Information

Supplemental information for this article can be found online at http://dx.doi.org/10.7717/peerj.16379#supplemental-information.

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
