# Peer review of "Light wavelength and pulsing frequency affect avoidance responses of Canada geese"

_PeerJ, doi:10.7717/peerj.16379_

## Round 0.1 · original submission · Minor Revisions

Thank you for submitting your manuscript to PeerJ. Your manuscript has been reviewed by two experts in the field, and they both agree that the manuscript is well-written and argued. Both reviewers offer some helpful and detailed suggestions for how to improve the presentation of your study. Please address each point made in your revised manuscript.

Additionally, I have a few suggestions/comments.

1) the uploaded document has tracked changes, in your revision please make sure you upload one clean copy and one with the tracked changes.

2) the figures have a lot of white space, which is usually fine, but here it does make things a little harder to read. Could you reformat them so the font is bigger, the images of the geese/setup in 1.2 are bigger and the datapoints and lines and font in figure 2 are bigger? Furthermore, the font size in Fig 2 across the panels is not consistent.

3) For Figure 2d, you have used the colours red and blue to indicate the light colours, but here you use blue and orange for a different meaning. This is confusing. Please choose other colours here.

4) For your linear and generalised linear mixed effect model selection process it would be helpful to report the next best-fit models and compare them with the full model and null model - such as reporting the AICs for each of the top 5 models. This will help the reader understand how much better your chosen best-fit model is. Your model selection explanation is a little confusing, and you don't report the results of that model selection process, so it is difficult for the reader to understand how you arrived at the model that you chose.

5) When you present your results if there is no significant difference then you do not need to report each trial average (e.g., line 581) because they are all statistically equivalent, nor should you report the direction of a trend (e.g., line 582-583) if there is no statistical relationship. Only report the effect size and direction of the factors that are statistically important. Because you spent a lot of time reporting and commenting on associations that were not statistically significant, it does make it hard for the reader to wade through and identify what were the important pieces that your research found. It looks like there were no significant associations in latency to respond, head movement or body movement, so what did you find? If you only found significant results in the probability of light avoidance, that is okay - just be clear and concise about what you found.

6) Your Table 1 includes the model results for the first 3 paragraphs of your results section, but not the results on "probability of light avoidance". It would be useful if you included those results as well. It might make the section a little easier to understand as there is a lot of text in the results section on that topic.

Overall, you have presented your study clearly and with some minor edits/adjustments I believe that it will be ready for publication. I look forward to reading your revised submission.

·

Basic reporting

no comment

Experimental design

no comment

Validity of the findings

no comment

Additional comments

The authors thoroughly documented the literature, methods used, data generated, and how their results fit into the larger context of what is known about avian vision and potential applications for reducing collisions. They also do a good job of addressing limitations of their experimental design and results.

While reading the manuscript, a couple of thoughts came to mind, but I do not necessarily expect the authors to address all of these in the paper given the scope of the study.
- Presumably, the geese in the study were only able to walk through the arena during the experiment. I'd be curious to know if there may be any expected differences in bird responses to light in flight (when they're more likely to collide with aircraft) vs when they are walking on the ground. I would imagine that latency to respond would be more important in flight.
- With the experiment being conducted outside of the migration season and during the day, as a reader, I'd be interested to know if the time of year and daytime vs nighttime conditions could impact bird responses to light stimuli (and by extension, how artificial light at night may impact responses for birds traveling at night/dawn/dusk). If I recall correctly, the FAA reports that most aircraft collisions occur during the day and from summer to fall. Perhaps some of this information could help place the experimental design in the context of real-world data.

I have attached an annotated document with just a few very minor suggestions for things I happened to notice, and I would like to commend the authors for their work.

Reviewer 2 ·

Basic reporting

Clear, unambiguous, professional English language used throughout.
Yes


Intro & background to show context. Literature well referenced & relevant.
Yes..


Structure conforms to PeerJ standards, discipline norm, or improved for clarity.
Manuscript contains PeerJ’s standard sections except for the Acknowledgments.


Figures are relevant, high quality, well labelled & described.
Yes


Raw data supplied.
Yes, authors provide as supplemental material multiple CSV files with data and an R script to replicate the analysis.

Experimental design

Original primary research within Scope of the journal.
The manuscript was sent out for review, then I guess yes.


Research question well defined, relevant & meaningful. It is stated how the research fills an identified knowledge gap.
Yes to all, but see my additional comments.


Rigorous investigation performed to a high technical & ethical standard.
Yes


Methods described with sufficient detail & information to replicate.
Yes, the methods are thoroughly described.

Validity of the findings

Impact and novelty not assessed. Meaningful replication encouraged where rationale & benefit to literature is clearly stated
Don´t know what you mean here.


All underlying data have been provided; they are robust, statistically sound, & controlled.
Yes to all.


Conclusions are well stated, linked to original research question & limited to supporting results.
Yes.

Additional comments

In this manuscript, authors analyzed whether Canada geese avoided or were attracted to two light colors (red and blue) under two different modes of operation: pulsing and steady. To test this, authors released geese into one end of an experimental enclosure and gave them the opportunity and motivation to walk towards the back end of the enclosure where geese decoys sat, having to choose whether to go through the left or right of a divide that also separated the light stimulus. Authors released geese into the enclosure one by one in repeated occasions throughout multiple days applying different treatment combinations (red/blue × pulsing/steady) through a randomized design. Authors gave up to 10 minutes to the goose to choose what side of the enclosure to walk to, and a trial was concluded once a goose’s beak crossed an imaginary line created by the divide of the enclosure, i.e. after a goose decided which side of the enclose to walk to.

The rationale behind these experiments seems to be to contribute with information about alternatives to reduce the risk of bird collisions with aircraft, with geese being an important subject given their size and the potential damage that these birds can inflict to aircraft.

This manuscript is well written and the Methods are thoroughly described. I don’t have that many suggestions but I wonder, however, 1) whether geese choosing to walk towards one side of an experimental enclosure with one particular color of light can be interpreted a geese exhibiting avoidance to the other color, and 2) whether these geese actually responded to the experimental treatments.

In the “Latency to respond” subsection of the Results, authors report the difference in seconds that geese took to respond to one or the other color of light. Put another way, the latency to respond also means the time a goose took to walk towards one or the other side of the divide. I think authors should also report the actual time goose took to choose and discuss their implications. Were the times short or long? Since authors gave geese up to 10 minutes to choose and they also state “If a choice was not made after 10 minutes the trial stopped” then my guess is the Latency to respond was on the long side. What does that imply? Is there anything in the literature documenting how fast does a reaction has to be to be considered an actual preference or avoidance response? Were your actual latency times within the time that geese or other birds take to avoid obstacles? There’s a report by the Ventana Wildlife Society (Ventana, 2009) documenting the reaction distance of birds to power lines. While the underlying data are not provided, we can deduce from their Figure 7 that at least some geese in flight reacted 10 – 25 m away from the power line. If any of those geese were flying at 40 mph or 65kph, the reaction time would have been seconds. Another example, more pertinent to this work and less restrictive about reaction distances, is that by Bellrose (1971) where they stated that: “We have noted, in over 2,000 hours of diurnal flying , that small birds such as blackbirds, swifts, and swallows, make last-second plunges to avoid aircraft, *but ducks and geese take evasive flight at much greater distances ahead of the aircraft*.” (text between asterisk is my way to highlighting here). I did not go to the depth of trying to find whether they reported those “greater distances”, but since they talk of birds seen in flight from a moving aircraft, I would still think that the reaction time would have been measured in seconds and not minutes. In this work, the distance from the release point to the divide was <5 meters, so geese were given10 minutes to walk 5 meters and not all of them did… that begs the question: did these geese actually respond to the treatments? Is there a possibility that these geese did not really chose one side of the divide based on the light treatment, but that they just walked in whatever direction for whatever other reason?

Given the apparent goal of contributing to the literature about aircraft safety, I think this manuscript would benefit with some discussion about bird reaction times to obstacles (especially aircraft or other moving vehicles), how those compare to what was recorded here, what these could mean in a real life situation when a moving aircraft approaches a flock of geese (or the other way around), whether the geese in these experiments actually responded to the treatment, and perhaps even to de-emphasize starting from the title that you documented avoidance responses.


Minor comments
Please consider adding a metadata file explaining what is contained in your supplemental material files, including definitions of the columns in the CSV files.

Please define JND.


References
• Bellrose, Frank C. “The Distribution of Nocturnal Migrants in the Air Space.” The Auk 88, no. 2 (1971): 397–424. https://doi.org/10.2307/4083887.

• Ventana Wildlife Society. (2009). Evaluating diverter effectiveness in reducing avian collisions with distribution lines at San Luis National Wildlife Refuge Complex, Merced County, California (CEC‐500‐2009‐078). California Energy Commission, Public Interest Energy Research (PIER) Program. http://citeseerx.ist.psu.edu/viewdoc/download?doi=10.1.1.386.2300&rep=rep1&type=pdf

---

## Round 0.2 · accepted · Accept

Congratulations! You have done an excellent job addressing the reviewers' comments - the manuscript is now ready for publication in PeerJ